# Screening of In Vitro Heavy Metal Tolerance in *Tocoyena brasiliensis* Mart. (Rubiaceae)

**DOI:** 10.3390/plants14091331

**Published:** 2025-04-28

**Authors:** Allex Sandro Durão Martins, Thais Huarancca Reyes, Lorenzo Guglielminetti, Cláudia Roberta Damiani

**Affiliations:** 1Faculty of Biological and Environmental Sciences, Federal University of Grande Dourados, Dourados 79804-970, MS, Brazil; allex.durao@gmail.com (A.S.D.M.); claudiadamiani@ufgd.edu.br (C.R.D.); 2Department of Agriculture, Food and Environment, University of Pisa, 56124 Pisa, Italy; thais.huarancca@agr.unipi.it

**Keywords:** Cerrado native species, survival rate, morphological traits, callus formation, zinc, lead, cadmium

## Abstract

Zinc (Zn: 0–400 mg L^−1^ zinc acetate), lead (Pb: 0–400 mg L^−1^ lead acetate), and cadmium (Cd: 0–8 mg L^−1^ cadmium chloride) tolerance in stem explants of *Tocoyena brasiliensis* Mart. from seeds collected in the Brazilian Cerrado were studied under controlled conditions. The explants showed a regular growth activity in a metal-free medium. All metals did not affect explant survival, except for 400 mg L^−1^ Zn, which resulted in lethality. Shoot number was not affected by metal treatment, while shoot length and leaf number varied depending on the metal. Cd induced a gradual reduction in leaf number without affecting shoot length. Pb gradually reduced the shoot length at concentrations beyond 200 mg L^−1^, while no effects were found in Zn concentrations from 0 to 200 mg L^−1^. Pb and Zn induced leaf production at 50 mg L^−1^, while a gradual reduction was observed with increasing concentration. Callus formation was not affected by Cd, while increasing Zn and Pb concentrations reduced this cell division and organization with Zn, showing drastic effects. Altogether, *T. brasiliensis* explants demonstrated high tolerance to Cd and Pb. However, further studies are needed to explore the phytoextraction capacity of this species at in vitro and planta levels.

## 1. Introduction

In recent years, the concentrations of several heavy metals in the environment have been increasing due to the heightened anthropogenic activities, excessive use of agrochemicals, and rapid industrialization. This accumulation is affecting soil and water quality, food security and animal and human health, thus becoming a serious problem [1]. In plants, some metals such as zinc (Zn) have been evolutionarily selected as essential elements, which, at adequate concentrations, are required for normal metabolic processes. Other metals that are not essential elements and without physiological functions, such as lead (Pb) and cadmium (Cd), are detrimental for plants even at low concentrations [2,3]. Heavy metals, whether essential or non-essential elements, in excess have similar harmful impacts, including reduced biomass production, growth and photosynthesis inhibition, chlorosis, altered fluid balance and nutrient absorption, as well as senescence, leading to plant death [4].

Phytotoxicity due to increased levels of heavy metals occurs through their reactions with sulfhydryl groups of functional proteins, metals’ affinity for reacting with phosphate groups and active groups of ADP or ATP, competition with and replacement of essential nutrient cations, and the production of reactive oxygen species that damage macromolecules [5]. Plants have developed different mechanisms of metal tolerance throughout evolution with strategies of metal uptake reduction, metal exclusion in specialized structures, chelation and cellular and subcellular compartmentalization [6].

The first line of heavy metals entry barriers occurs in the root, where their uptake is affected by factors such as metal ion mobility, bioavailability, root organic content, root exudates, and rhizosphere microbial composition [7]. Ion metals can enter into root tissues via apoplast and symplast movements. This allows their translocation to aboveground parts of the plant for immobilization into the vacuoles, rendering heavy metals inactive and thus preventing damage to essential metabolic processes [5]. Moreover, during metal stress, the synthesis of stress-related proteins and hormones, antioxidants, and signaling molecules is induced to avoid morphological, metabolic, and physiological abnormalities in plants. It has been reported that metal resistance and related defense mechanisms are plant species-, clone-, and even hybrid-specific, and they are also dependent on the type, concentration, and oxidation state of heavy metals [7,8].

Identifying suitable plant species that tolerate heavy metal contamination is essential for testing the remediation capacities of polluted soils and agricultural lands. Metal uptake by woody species can be more effective in comparison to herbaceous types; however, their cultivation is inherently time consuming. In this context, the development and exploitation of different model systems, such as in vitro culture, have proven to be useful tools [8]. Indeed, in vitro screening represents a quick way to initially characterize plant metal tolerance by dissecting the plant–soil–microorganism system to evaluate the stress factor and by reducing the period of growth and treatment in new species that require big spaces [9].

In Brazil, the Cerrado region is located in the central part of the country, consisting of about 205 million hectares, which is equivalent to 23% of the territory [10]. The soil in this region is characterized by two dominant classes: Latosol (44.1%) and Quartzarenic Neosol (21.4%) [11]. These soils have low natural fertility, low pH (4.5–5.5), high aluminum saturation, and high phosphorus fixation capacity [10,12]. Despite the nutrient deficiency and toxicity in Cerrado soils, numerous native species are resistant or tolerant to these conditions that are considered unfavorable to cultivated plants [13]. For instance, a recent study showed that trees in the Cerrado biome, such as *Copaifera langsdorffii* Desf., *Cedrella fissilis* Vell., and *Handroanthus impetiginosus* (Mart. ex DC.) Mattos, had good tolerance to metal contamination, suggesting their possible applications in remediation [14].

*Tocoyena brasiliensis* Mart., popularly known as jenipapinho or jenipapo-bravo, is a woody species from the Rubiaceae family that is predominately found in the Cerrado biome [15]. Few studies have focused on this species, particularly on the activity of its secondary metabolites, such as the antifungal activity of triterpenes from leaf extract against *Cladosporium cladosporioides* strains [16]. Although studies related to its tolerance to contaminants such as heavy metals have not been found in the literature so far, recent studies have shown that this species, as well as others in the Cerrado region, are adapted to poor soils with a natural tendency to be used in degraded areas [14,17]. The ability of *T. brasiliensis* to thrive in acidic and metal-rich soils led us to hypothesize that this woody species might evolve robust metal tolerance mechanisms. Therefore, the aim of this study was to establish, for the first time, the in vitro growth conditions of *T. brasiliensis* stem explants under a controlled environment, and to evaluate the explants’ metal tolerance toward Zn, Pb, and Cd separately by applying the in vitro system. This initial screening highlighted the potential of the studied species; however, it is advisable to confirm the data in this study through full plant culture and field performance trials.

## 2. Results

### 2.1. In Vitro Culture Establishment

The explants cultured on control woody plant medium (WPM) (0 mg L^−1^) without growth regulators showed a regular growth activity (Figure 1), indicating that the used protocol was suitable for the in vitro culture of *T. brasiliensis*. Then, the metal tolerance of this woody species was screened in vitro, as it is a quick method avoiding the soil matrix interferences. Here, explants exposed to Zn, Pb and Cd, separately, showed some visual symptoms of toxicity, such as leaf necrosis and chlorosis, depending on the metal and concentration. In detail, Zn at concentrations ≥ 100 mg L^−1^ induced paleness, yellowing, and chlorosis, resulting in necrosis when explants were exposed to 400 mg L^−1^ (Figure 1A). Differently, Pb up to 200 mg L^−1^ did not result in an apparent phytotoxicity as explants showed green leaves, while chlorosis and few necrotic symptoms were observed at 400 mg L^−1^ (Figure 1B). Explants exposed to >4 mg L^−1^ Cd resulted in leaf chlorosis and wilting, while those grown at 8 mg L^−1^ also showed some necrotic signs in leaves (Figure 1C).

The survival percentage of explants represented the visual symptoms’ quantification of phytotoxicity (Figure 2). The results showed that Zn treatment significantly affected (*p* < 0.0001; Appendix A) and negatively correlated (r = −0.88; Appendix A) with explant survival. No significant differences were found between the control (0 mg L^−1^) and explants grown up to 200 mg L^−1^ Zn, while those exposed to 400 mg L^−1^ could not survive (Figure 2A). Differently, Pb and Cd treatments did not significantly affect (*p* > 0.05; Appendix A; Figure 2B,C) and showed a low and negative correlation with the survival of *T. brasiliensis* (r = −0.36 and r = −0.43; Appendix A).

### 2.2. Growth-Related Parameters

Morphological analysis in control and treated plants showed that different concentrations of Zn (Figure 3A), Pb (Figure 3B), and Cd (Figure 3C) did not result in significant variations in shoot number (Appendix A). Moreover, low correlations were found between these metals and the shoot number (Appendix A). Differently, the length was significantly affected by Zn and Pb but not by Cd (Appendix A; Figure 4C), and it showed a strong and negative correlation only with Zn (r = −0.88; Appendix A). It was noticed that shoot length was not significantly altered with the increase in Zn concentration up to 200 mg L^−1^, which was followed by an abrupt reduction at the highest concentration (400 mg L^−1^; Figure 4A). Although the results related to Pb showed a similar quadratic pattern as Zn, some differences were observed (Figure 4B). In detail, explants grown at 50 and 100 mg L^−1^ Pb showed similar shoot lengths to those grown under control condition (i.e., Pb-free culture), which was followed by a gradual decrease with higher concentrations (i.e., 200 and 400 mg L^−1^).

Plants cultured in different concentrations of Zn, Pb, and Cd showed significant changes in leaf number per shoot (Appendix A). However, the strongest and most inverse relationship was detected between Zn and leaf number (r = −0.91; Appendix A). For Zn, the leaf number increased in plants grown from 0 to 50 mg L^−1^, which was followed by a gradual reduction, ultimately resulting in no leaves under the highest Zn concentration (400 mg L^−1^; Figure 5A). Similarly, explants exposed to Pb increased the leaf number when concentrations changed from 0 to 50 mg L^−1^ (Figure 5B). This was followed by a slight reduction up to 200 mg L^−1^ and a subsequent significant decrease in leaves at 400 mg L^−1^ (Figure 5B). Unlike Zn and Pb, the leaf number per shoot showed a consistent decrease in response to Cd with a gradual reduction as the metal concentration increased (Figure 5C).

### 2.3. Callus Formation

Different concentrations of Zn and Pb resulted in significant variations in callus formation, while Cd did not (Appendix A; Figure 6C). Similarly, callus growth showed a strong and negative correlation with Zn and Pb (r = −0.71 and r = −0.96; Appendix A), while a low correlation was detected with Cd (r = −0.26; Appendix A). Our control condition (0 mg L^−1^) induced the generation of callus after 30 days of culture (Figure 6). However, Zn negatively affected this induction, showing an abrupt decrease even at the lowest concentration tested (i.e., 50 mg L^−1^) and completely repressing callus formation when explants were grown at concentrations ≥ 200 mg L^−1^ (Figure 6A). Pb caused a gradual reduction in callus growth with increasing concentrations from 0 to 100 mg L^−1^ with no significant differences compared to the control condition (Figure 6B). This negative trend was significantly accentuated at 200 mg L^−1^, which was followed by complete repression at the highest concentration tested (400 mg L^−1^; Figure 6B).

## 3. Discussion

This study describes the effects of different concentrations of essential and non-essential metals on some biometric traits in *T. brasiliensis* explants using an in vitro model system to screen the metal tolerance of this not yet studied woody species. However, since a common tolerance strategy in plants includes restricting metals to the roots, it is advisable to test our results in shoots with roots.

### 3.1. Explants Sensitivity to High Concentration of Zn

Almost all biometric parameters were affected by Zn concentration, causing chlorosis and yellowing in a concentration-dependent manner, becoming lethal at the highest concentration (i.e., 400 mg L^−1^ = 2.2 mM). Zn is an essential trace metal that, among other micronutrients, is needed to support plant growth and development. For instance, Zn is a cofactor of many enzymes and transcription factors involved in basic plant functions such as phenol metabolism, starch formation, and the increase in cellular size and differentiation [18]. However, at elevated concentrations, it becomes toxic due to its negative effect on chlorophyll biosynthesis in the leaf and carbon assimilation [19,20]. Moreover, Zn negatively affects nutrient balance and induces oxidative stress, altering a range of interactions at the molecular and cellular levels [21,22,23]. Concordantly, these alterations might occur in explants grown at 400 mg L^−1^ Zn, leading to their death.

The observed toxic symptoms at Zn concentrations ≥ 200 mg L^−1^ (=1.1 mM) may be caused by chloroplast damage, decreasing chlorophyll content and subsequent chlorosis [22]. Zn can compete with Mg altering chlorophyll structure, and it can also affect the expression of genes related to chlorophyll biosynthesis [19,24]. The damage of chloroplasts can be also due to oxidative stress, as excess Zn leads to the production of reactive oxygen species [25]. A previous study showed that Zn phytotoxicity in poplar happened at 1 mM due to the Zn accumulation in aerial parts [19]. Moreover, Zn uptake to shoots and chlorosis symptoms was also observed in rooted and rootless shoots using in vitro systems [26]. It would be interesting to evaluate whether *T. brasiliensis* uptakes Zn and whether the observed phytotoxicity also occurs in rooted plants.

The fact that Zn negatively affected callus formation could be related to the alteration of auxin-induced callus [27]. Indeed, Zn in excess can negatively affect the synthesis of auxin by modulating key auxin biosynthesis genes [28]. The effect of Zn on other hormones involved in callus formation is not excluded [29]. Moreover, Wang et al. [28] showed that Zn inhibited root growth by interfering with the auxin accumulation in a dose-dependent manner. The reduction in plant growth was also observed in poplar [19,26] and in other woody species in the Cerrado biome [22]. This suggests that the high concentrations of Zn used in this study could alter hormone homeostasis, which in turn may lead to a reduction in leaf number per shoot. Further studies are needed to reveal hormone alterations in *T. brasiliensis* by increasing Zn concentrations.

### 3.2. Explants Resistance to Cd

Although Cd and Zn have similar mineralogical origins and chemical characteristics [30], Cd is not an essential element and is highly toxic to living organisms, even at low concentrations [23]. In this study, plant survival was not affected by Cd concentrations, highlighting *T. brasiliensis* explants’ tolerance to this heavy metal. However, Cd concentrations ≥ 6 mg L^−1^ (=33 µM) reduced the leaf number per shoot. Concordantly, Souza et al. [31] found that in another woody species of the Rubiaceae family in the Cerrado biome, Cd increased the lignification process in root and leaf tissues specifically at concentrations ≥ 4 mg L^−1^ (=17.5 µM). This anatomical change reduces the cell wall plasticity, affecting plant growth and development. Moreover, a recent study demonstrated that plant tolerance to Cd stress occurs by regulating gene expressions involved in cell wall synthesis [32]. Further studies are needed to verify whether the reduction in leaves in *T. brasiliensis* in response to Cd is related to the lignification process.

Here, symptoms of leaf yellowing were also observed at concentrations ≥ 6 mg L^−1^ (= 33 µM). This phenotype may be related to Cd-induced chlorophyll alterations. Concordantly, previous studies demonstrated that Cd stress inhibits chlorophyll synthesis with a deleterious effect on synthesizing enzymes [33,34]. Moreover, Grajek et al. [35] demonstrated that Cd can replace Mg in the chlorophyll molecule, inducing its degradation. These changes in chlorophyll lead to photosynthesis inhibition. Indeed, previous studies demonstrated that high concentrations of Cd reduce photosynthesis [31,36]. Moreover, Andrade et al. [37] showed that the reduction in leaf number was a direct consequence of the reduced photosynthetic capacity, affecting plant growth and biomass production. This last effect may be another reason, besides the lignification process, why the leaf number in *T. brasiliensis* decreased in response to Cd stress. It would be important to evaluate the physiological alterations at the photosynthetical level in response to Cd.

Cd also induces reactive oxygen species, which can result in oxidative injuries when cellular redox balance favors pro-oxidants [38]. However, plants produce antioxidants to scavenge the excessive reactive oxygen species, reducing injuries [38]. Here, visible injuries in the explants were not observed, suggesting that *T. brasiliensis* may maintain the balance between antioxidants and pro-oxidants. However, more in-depth studies are needed to validate this speculation.

### 3.3. Explants Resistance to Pb

Similar to Cd, *T. brasiliensis* explants survival was not affected by Pb, indicating that this species is more tolerant to Cd and Pb than to Zn even though the former metals are considered the most hazardous due to their persistence in the environment [39]. Despite explants’ resistance to Pb, chlorosis and few necrotic signs were observed at the highest concentration. Interestingly, the initial symptoms of necrosis in Pb-exposed explants were registered at similar concentrations to those grown in Zn-containing medium (i.e., 400 mg L^−1^ = 1.2 mM and 200 mg L^−1^ = 1.1 mM for Pb and Zn, respectively). Indeed, it has been demonstrated that essential and non-essential metals generally produce common toxic effects that ultimately can cause cell death [5].

Pb at high concentrations affects the biosynthesis of chlorophyll, carotenoids, and plastoquinone, as well as reduces the activity of C_3_ cycle enzymes, thereby hampering photosynthesis [40]. Pb can substitute Mg, altering chlorophyll synthesis and inhibiting enzymatic activity and electron transport in the Calvin cycle [40,41]. These effects may be related to the observed leaf chlorosis in *T. brasiliensis* exposed to 400 mg L^−1^ Pb. Moreover, Pb in excess also accelerates the production of reactive oxygen species, which damage macromolecules and affect many cellular processes, leading to cell death [42]. This generally happens when scavengers cannot be sustained at certain levels to overcome oxidative damage. Thus, it is plausible to hypothesize that 400 mg L^−1^ Pb may induce an imbalance between antioxidants and pro-oxidants, causing necrotic lesions, while lower concentrations may not alter the redox balance in *T. brasiliensis*. In addition, these biochemical responses may result in growth inhibition (e.g., leaf number and shoot length). However, more studies are needed to determine biochemical process alterations in explants in response to increasing Pb concentration and their relationship with plant growth impairment.

Here, explants showed particular morphological changes with increasing Pb concentrations in comparison with the control. A transient increase in leaf number was reported in explants at 50 mg L^−1^ Pb (0.15 mM). Similarly, aspen cultures exposed to 0.1 mM Pb, epazote explants to 4.5–7.5 µM Pb, and duckweed plants to low Pb concentration (2.6 µM) did not show toxicity signs, and plant growth was even promoted [43,44,45]. When reactive oxygen species production does not exceed antioxidants scavenger potential, these can act as signaling molecules involved in various physiological process, including growth by the regulation of plant hormones [5]. In this context, it has been demonstrated that depending on the concentration, Pb can initiate signaling cascades modulating phytohormones that in turn might affect leaf number [46,47]. Hormonal changes in relation to leaf production in *T. brasiliensis* exposed to 50 mg L^−1^ Pb need to be clarified.

Another biometric trait that was affected by Pb was the formation of calluses, which decreased with increasing metal concentration, especially at concentrations beyond 100 mg L^−1^. Accordingly, Amdoun et al. [48] found that Pb had inhibitory effects on callus formation in another woody species. Thin inhibitory effect may be related to the fact that Pb can cause alterations in plant metabolism, modifying the hormonal balance and disturbing cell formation [49].

## 4. Materials and Methods

### 4.1. Plant Material and Growth Conditions

The seeds of *Tocoyena brasiliensis* Mart. (Rubiaceae) were obtained from fruits collected in the Assentamento Lagoa Grande, located in the district of Itahum, municipality of Dourados—MS, Brazil (21°59′43″ S, 55°19′38.7″ W). The experiments were carried out in the Plant Biotechnology Laboratory, Faculty of Biological and Environmental Sciences, Federal University of Grande Dourados, Brazil.

In a laminar flow chamber, the seeds were superficially sterilized by immersion and constant agitation in different disinfectant solutions as follows: 5 min in sodium hydroxide solution (0.1 M), 1 min in 70% alcohol, and 15 min in sodium hypochlorite (2.5%) with Tween 80 detergent (0.5 mL L^−1^). After surface sterilization, seeds were washed in sterilized water (3×) and sowed in solid WPM medium [50]. Cultures were incubated in a growth chamber in the dark at controlled temperature (25 ± 2 °C) for 7 days and then shifted to 100 µmol m^−2^ s^−1^ photosynthetic active radiation (PAR) with a 16 h photoperiod. After germination, the plantlets obtained were multiplied in vitro in solid WPM medium at controlled temperature (25 ± 2 °C), 100 µmol m^−2^ s^−1^ PAR with a 16 h photoperiod.

### 4.2. Heavy Metal Treatments

Under sterilized conditions, nodal stem explants with two lateral buds and leaves were subcultured into clear glass jars (65 × 90 mm) with plastic lids, containing 40 mL WPM medium supplemented with varying concentrations of zinc (Zn), lead (Pb) or cadmium (Cd) along with 100 mg L^−1^ myo-inositol, 30 g L^−1^ sucrose, and 60 g L^−1^ agar at pH 5.8. WPM without growth regulators was used because in a previous study, explant growth did not show differences when cultivated in WPM with or without different growth regulators. The increasing metal concentrations used in this study were 0, 50, 100, 200, and 400 mg L^−1^ zinc acetate for Zn; 0, 50, 100, 200, and 400 mg L^−1^ lead acetate for Pb; and 0, 2, 4, 6, and 8 mg L^−1^ cadmium chloride for Cd. Free-metal containing medium is referred to as the control. The tested concentrations for Cd and Pb were in accordance with the CONAMA resolution 420/2009 [51], which establishes criteria and guiding values for soil quality regarding the presence of chemical substances and sets guidelines for the environmental management of areas contaminated by these metals as a result of anthropogenic activities in Brazil. The tested concentrations for Zn were based on a study conducted by Fageria [52].

Each jar contained five explants (each treatment consisted of five replicates), and the growth condition was set at 25 ± 2 °C, 100 µmol m^−2^ s^−1^ PAR, and 16 h photoperiod for a period of 30 days. All chemicals were of analytical reagent grade.

### 4.3. Data Collection

At 30 days of treatment, the following variables were evaluated: number of leaves per shoot, number of shoots, shoot length, callus formation at the base of shoot explants, and survival (as maintenance of green color). For explants that produced shoots but died throughout the experiment, the variables of leaf number, shoot length and callus formation were not measured and set as zero.

### 4.4. Statistical Analysis

All experiments were conducted in a completely randomized experimental design with five replicates. The obtained data were subjected to one-way analysis of variance (ANOVA). Significant differences among means were estimated at the level of *p* < 0.05 by using Tukey test. The assumption of normality was assessed using the Shapiro–Wilks test. Statistical evaluation of the data was performed with GraphPad Prism version 10.0.0 (GraphPad Software, San Diego, CA, USA). Correlation analysis was used to assess associations between each metal and the biometric parameters.

## 5. Conclusions

This study established, for the first time, the in vitro growth conditions of *T. brasiliensis* stem explants under a controlled environment. This model system provided a fast way to screen the tolerance and survival capacity of this woody species toward Zn, Pb, and Cd. Zn concentrations beyond 200 mg L^−1^ were toxic for explants, causing lethality. In contrast, explants could survive Pb and Cd even at high concentrations (i.e., 400 and 8 mg L^−1^, respectively), indicating high tolerance to these metals, although some morphological effects were observed in a metal-dependent manner. Altogether, *T. brasiliensis* seems to be a promising species to further explore its phytoextraction capacity and defense mechanisms in full plant culture system and field performance trials.

## Figures and Tables

**Figure 1 plants-14-01331-f001:**
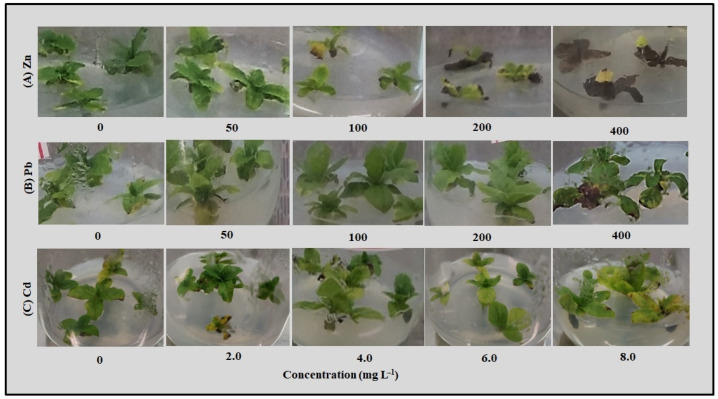
Phenotypical picture of *T. brasiliensis* after metal treatment in solid culture. Explants were grown in WPM medium containing varying concentrations (mg L^−1^) of (**A**) Zn, (**B**) Pb, and (**C**) Cd. Pictures were taken at 30 days of treatment, and representative ones are shown.

**Figure 2 plants-14-01331-f002:**
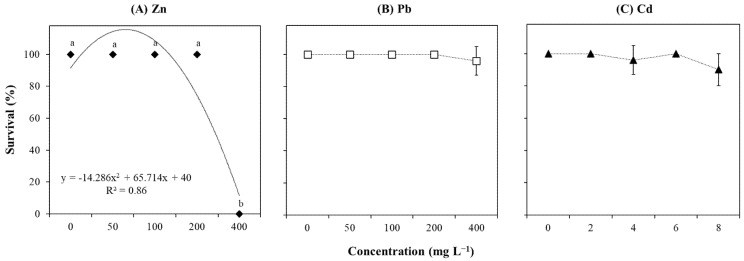
Survival (%) of *T. brasiliensis* after metal treatment in solid culture. Explants were grown in WPM medium containing varying concentrations (mg L^−1^) of (**A**) Zn, (**B**) Pb, and (**C**) Cd. Trendlines with their respective equations and R-squared values are shown when available. Error bars represent the standard deviation of the mean (n = 5). One-way ANOVA was performed to determine the influence of treatment on explant survival (details in Appendix A). Different lowercase letters indicate significant differences between the means (*p*  <  0.05) according to Tukey’s test.

**Figure 3 plants-14-01331-f003:**
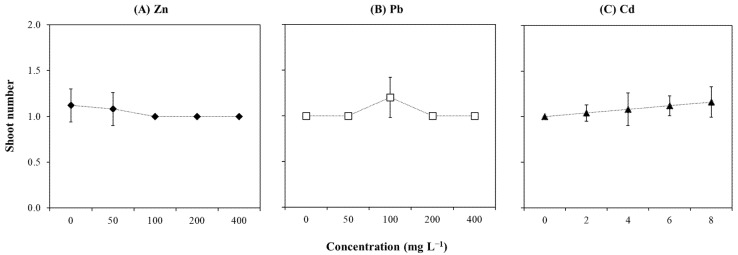
Shoot number of *T. brasiliensis* after metal treatment in solid culture. Explants were grown in WPM medium containing varying concentrations (mg L^−1^) of (**A**) Zn, (**B**) Pb, and (**C**) Cd. Error bars represent the standard deviation of the mean (n = 5). One-way ANOVA was performed to determine the influence of treatment on shoot number (details in Appendix A).

**Figure 4 plants-14-01331-f004:**
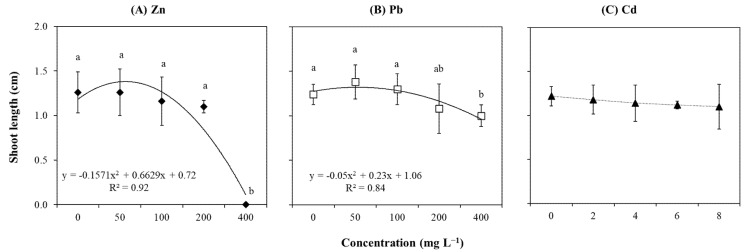
Shoot length (cm) of *T. brasiliensis* after metal treatment in solid culture. Explants were grown in WPM medium containing varying concentrations (mg L^−1^) of (**A**) Zn, (**B**) Pb, and (**C**) Cd. Trendlines with their respective equations and R-squared values are shown when available. Error bars represent the standard deviation of the mean (n = 5). One-way ANOVA was performed to determine the influence of treatment on shoot length (details in Appendix A). Different lowercase letters indicate significant differences between the means (*p*  <  0.05) according to Tukey’s test.

**Figure 5 plants-14-01331-f005:**
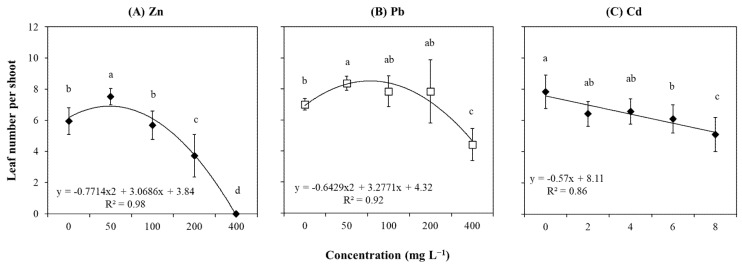
Leaf number per shoot in *T. brasiliensis* after metal treatment in solid culture. Explants were grown in WPM medium containing varying concentrations (mg L^−1^) of (**A**) Zn, (**B**) Pb, and (**C**) Cd. Trendlines with their respective equations and R-squared values are shown. Error bars represent the standard deviation of the mean (n = 5). One-way ANOVA was performed to determine the influence of treatment on leaf number per shoot (details in Appendix A). Different lowercase letters indicate significant differences between the means (*p*  <  0.05) according to Tukey’s test.

**Figure 6 plants-14-01331-f006:**
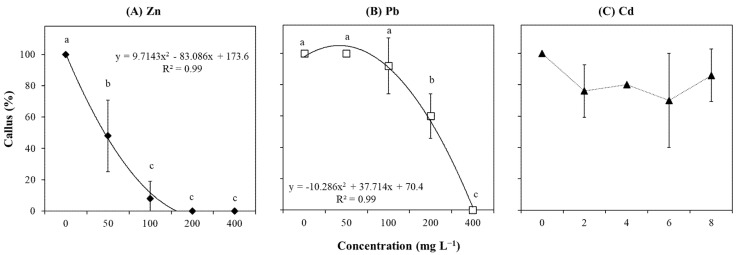
Callus formation (%) of *T. brasiliensis* after metal treatment in solid culture. Explants were grown in WPM medium containing varying concentrations (mg L^−1^) of (**A**) Zn, (**B**) Pb, and (**C**) Cd. Trendlines with their respective equations and R-squared values are shown when available. Error bars represent the standard deviation of the mean (n = 5). One-way ANOVA was performed to determine the influence of treatment on callus formation (details in Appendix A). Different lowercase letters indicate significant differences between the means (*p*  <  0.05) according to Tukey’s test.

## Data Availability

Data are contained within the article.

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
