# Peer review of "Screening of In Vitro Heavy Metal Tolerance in Tocoyena brasiliensis Mart. (Rubiaceae)"

_plants, 2025, doi:10.3390/plants14091331_

Round 1
Reviewer 1 Report
Comments and Suggestions for Authors
The work has a potential foundation for an article. If you want to certify the use of this species for phytoremediation, you must additionally provide information on the accumulation of the heavy metals investigated in shoots and roots. In your experiment, you examined the plant's tolerance to various concentrations of HMs using explants without roots. The data is really intriguing, but the phytoremediation potential examined differs from using the full plant. My advice is to get this information and incorporate it into this work. If the seeds are from various plants, it is also possible to screen clones with varying capacities for uptake or resistance to the HMs utilized; in vitro propagation is an excellent tool for screening, selecting, and propagating the best candidates.
Anyway, some suggestion regarding the submitted article:
67 Result and discussion need to be in two different sections.
My suggestion is to move like supplementary data table 1, 2 and 3. I will also move at the end of the result description the paragraph from 68 to 73
Try to insert some sub-paragraph to the result and discussion sections.
Figure 1, 2,3,4 ,5 without standard deviation.
Author Response
Reviewer 1:
The work has a potential foundation for an article. If you want to certify the use of this species for phytoremediation, you must additionally provide information on the accumulation of the heavy metals investigated in shoots and roots. In your experiment, you examined the plant's tolerance to various concentrations of HMs using explants without roots. The data is really intriguing, but the phytoremediation potential examined differs from using the full plant. My advice is to get this information and incorporate it into this work. If the seeds are from various plants, it is also possible to screen clones with varying capacities for uptake or resistance to the HMs utilized; in vitro propagation is an excellent tool for screening, selecting, and propagating the best candidates.
Reply: Thank you for your insightful comments and suggestions. We have revised the manuscript following your suggestions and focusing on the plant's tolerance to various concentrations of heavy metals. Currently, we are conducting further experiments in planta to confirm the data obtained from our explants. We understand that using the full plant is crucial for accurately assessing phytoremediation potential. However, please note that woody species require a longer growth period.
Anyway, some suggestion regarding the submitted article:
- 67 Result and discussion need to be in two different sections.
Reply: The item in question has been divided as suggested by the reviewer.
- My suggestion is to move like supplementary data table 1, 2 and 3. I will also move at the end of the result description the paragraph from 68 to 73
Reply: The one-way ANOVA has been included as supplementary data, as suggested by the reviewer.
- Try to insert some sub-paragraph to the result and discussion sections.
Reply: Sub-paragraphs have been included as suggested by the reviewer.
- Figure 1, 2,3,4 ,5 without standard deviation.
Reply: Standard deviations have been added to all figures in the revised draft.
Reviewer 2 Report
Comments and Suggestions for Authors
The manuscript entitled “Screening of In Vitro Heavy Metal Tolerance in Tocoyena brasiliensis Mart. (Rubiaceae): Unlocking Possible Remediation Application” investigates the in vitro tolerance of Tocoyena brasiliensis stem explants cultivated in a woody plant medium containing different concentrations of zinc (Zn), lead (Pb), and cadmium (Cd). The study aims to evaluate the potential of this species for heavy metal remediation.
Overall Presentation of the Paper: The manuscript falls significantly below the expected standard for publication in Plants. The introduction is poorly structured, lacks coherence, and fails to establish the novelty of the study. The objectives are vague, and the hypothesis is completely missing. The results and discussion are improperly merged, creating confusion. Data presentation is extremely weak, with inadequate statistical analyses and subpar figures and tables that do not effectively support the study’s claims. Hence, the manuscript is not suitable for publication in its current form. It requires extensive revisions in nearly all sections, including the introduction, methodology, data presentation, discussion, and language. Major restructuring is needed to meet the standards of a high-impact journal.
General Comments:
- The introduction is written in an unstructured and disorganized manner. It lacks logical flow and fails to demonstrate the study's novelty.
- The manuscript does not clearly define its research objectives. There is no hypothesis, making it appear directionless.
- Combining results and discussion is an unacceptable approach. This format significantly reduces readability and makes it hard to extract meaningful insights.
- Data presentation is highly inadequate, with substandard tables and figures that lack clarity and proper labeling.
- The discussion is superficial and fails to provide an in-depth comparison with existing literature. It reads more like a repetition of results rather than a critical analysis.
- Language quality is poor, with awkward phrasing and grammatical errors throughout the manuscript. The manuscript needs extensive language editing.
Specific Comments:
Introduction:
- The introduction lacks structure and clarity. The background information is fragmented and does not logically lead to the research problem.
- There is no clear justification for conducting this study. The authors fail to highlight the significance of their work in the broader context of phytoremediation.
- The study’s objectives are unclear and scattered throughout the introduction instead of being explicitly stated at the end.
- The research hypothesis is completely absent. Without a hypothesis, the study appears aimless.
- Many citations are outdated, and key recent studies in the field are missing. The literature review is weak and lacks depth.
Materials and Methods:
- The methodology lacks critical details. The selection of metal concentrations is arbitrary and not justified.
- Statistical analysis is not well explained. The manuscript does not specify the statistical tests used to validate the findings.
- The description of experimental design is incomplete. More information is needed about replication and controls.
Results and Discussion:
- Results and discussion should be presented separately. The current format is confusing and does not allow a clear interpretation of the findings.
- Figures and tables are poorly presented, with minimal explanatory detail. They do not sufficiently support the claims made in the text.
- The discussion lacks scientific depth and does not critically engage with relevant literature. It mostly reiterates results without meaningful interpretation.
- The authors make broad claims without sufficient experimental evidence. This weakens the credibility of the study.
Conclusion:
- The conclusion is vague and lacks any substantial takeaway message.
- No clear recommendations for future research are provided. The authors need to discuss the practical implications and limitations of their study.
- The final statements are generic and do not provide a strong closure to the manuscript.
Extensive english editing is required.
Author Response
Reviewer 2:
The manuscript entitled “Screening of In Vitro Heavy Metal Tolerance in Tocoyena brasiliensis Mart. (Rubiaceae): Unlocking Possible Remediation Application” investigates the in vitro tolerance of Tocoyena brasiliensis stem explants cultivated in a woody plant medium containing different concentrations of zinc (Zn), lead (Pb), and cadmium (Cd). The study aims to evaluate the potential of this species for heavy metal remediation.
Overall Presentation of the Paper: The manuscript falls significantly below the expected standard for publication in Plants. The introduction is poorly structured, lacks coherence, and fails to establish the novelty of the study. The objectives are vague, and the hypothesis is completely missing. The results and discussion are improperly merged, creating confusion. Data presentation is extremely weak, with inadequate statistical analyses and subpar figures and tables that do not effectively support the study’s claims. Hence, the manuscript is not suitable for publication in its current form. It requires extensive revisions in nearly all sections, including the introduction, methodology, data presentation, discussion, and language. Major restructuring is needed to meet the standards of a high-impact journal.
Reply: Thank you for your thorough review and constructive criticism. We have carefully revised the manuscript, addressing all the points raised by the reviewer.
General Comments:
- The introduction is written in an unstructured and disorganized manner. It lacks logical flow and fails to demonstrate the study's novelty.
- The manuscript does not clearly define its research objectives. There is no hypothesis, making it appear directionless.
- Combining results and discussion is an unacceptable approach. This format significantly reduces readability and makes it hard to extract meaningful insights.
- Data presentation is highly inadequate, with substandard tables and figures that lack clarity and proper labeling.
- The discussion is superficial and fails to provide an in-depth comparison with existing literature. It reads more like a repetition of results rather than a critical analysis.
- Language quality is poor, with awkward phrasing and grammatical errors throughout the manuscript. The manuscript needs extensive language editing.
Reply: We have thoroughly revised the manuscript to improve the structure, define clear objectives and hypothesis, separate results and discussion, enhance data presentation, provide a more in-depth discussion, and correct language issues.
Specific Comments:
Introduction:
- The introduction lacks structure and clarity. The background information is fragmented and does not logically lead to the research problem.
- There is no clear justification for conducting this study. The authors fail to highlight the significance of their work in the broader context of phytoremediation.
- The study’s objectives are unclear and scattered throughout the introduction instead of being explicitly stated at the end.
- The research hypothesis is completely absent. Without a hypothesis, the study appears aimless.
- Many citations are outdated, and key recent studies in the field are missing. The literature review is weak and lacks depth.
Reply: The introduction has been modified following the detailed feedback of the reviewer.
Materials and Methods:
- The methodology lacks critical details. The selection of metal concentrations is arbitrary and not justified.
- Statistical analysis is not well explained. The manuscript does not specify the statistical tests used to validate the findings.
- The description of experimental design is incomplete. More information is needed about replication and controls.
Reply: The materials and methods have been revised as suggested.
Results and Discussion:
- Results and discussion should be presented separately. The current format is confusing and does not allow a clear interpretation of the findings.
- Figures and tables are poorly presented, with minimal explanatory detail. They do not sufficiently support the claims made in the text.
- The discussion lacks scientific depth and does not critically engage with relevant literature. It mostly reiterates results without meaningful interpretation.
- The authors make broad claims without sufficient experimental evidence. This weakens the credibility of the study.
Reply: The results and discussion sections have been separated, figures and tables revised, and the discussion critically rewritten as suggested.
Conclusion:
- The conclusion is vague and lacks any substantial takeaway message.
- No clear recommendations for future research are provided. The authors need to discuss the practical implications and limitations of their study.
- The final statements are generic and do not provide a strong closure to the manuscript.
Reply: The conclusion has been revised, taking into consideration reviewer’s comments.
Reviewer 3 Report
Comments and Suggestions for Authors
In the present manuscript, “screening of in vitro heavy metal tolerance in Tocoyena brasiliensis Mart. (Rubiaceae): unlocking possible remediation application” the authors evaluated the in vitro tolerance of T. brasiliensis stem explants cultivated in woody plant medium containing different concentrations of Zn, Pb and Cd. The manuscript is written well, however, it still needs substantial revisions before publication.
What were the concentrations of Zn, Pb, and Cd used? This information must be added to the abstract section.
The abstract section must be revised and contain the specific and quantitive data of study.
Choose expressive and significant keywords without repeating any words from the manuscript title.
There are numerous typos, and many passages are difficult to follow. A thorough revision is necessary to improve clarity and coherence starting from the abstract to the conclusion section.
What are the toxic impacts of Zn, Pb, and Cd on plants? The authors must add this information in an updated version of the manuscript.
I did not see any study hypothesis. Therefore, I suggest the authors please add the study hypothesis in the updated manuscript.
The objectives are missing from the introduction section. Please add the clear objective of your study in the updated version.
The results and discussion section should be separate. Please improve the discussion section, add strong logical reasoning, and compare the results with previous studies.
The authors have collected only limited data, which is not good enough to publish in high-quality journals like Plants. I suggest adding more data on physiological and biochemical traits.
The conclusions are not quantitative and should be rewritten for clarity, conciseness, and improved English.
Comments on the Quality of English LanguageMinor changes are needed.
Author Response
Reviewer 3:
In the present manuscript, “screening of in vitro heavy metal tolerance in Tocoyena brasiliensis Mart. (Rubiaceae): unlocking possible remediation application” the authors evaluated the in vitro tolerance of T. brasiliensis stem explants cultivated in woody plant medium containing different concentrations of Zn, Pb and Cd. The manuscript is written well, however, it still needs substantial revisions before publication.
- What were the concentrations of Zn, Pb, and Cd used? This information must be added to the abstract section.
Reply: The required information has been added to the abstract section.
- The abstract section must be revised and contain the specific and quantitive data of study.
Reply: The abstract has been carefully revised and rewritten as suggested.
- Choose expressive and significant keywords without repeating any words from the manuscript title.
Reply: Keywords have been modified as suggested.
- There are numerous typos, and many passages are difficult to follow. A thorough revision is necessary to improve clarity and coherence starting from the abstract to the conclusion section.
Reply: The manuscript has been carefully revised and modified.
- What are the toxic impacts of Zn, Pb, and Cd on plants? The authors must add this information in an updated version of the manuscript.
Reply: The required information has been added to the revised manuscript.
- I did not see any study hypothesis. Therefore, I suggest the authors please add the study hypothesis in the updated manuscript.
Reply: The hypothesis has been added in the revised manuscript.
- The objectives are missing from the introduction section. Please add the clear objective of your study in the updated version.
Reply: The objectives have been clearly added in the revised manuscript.
- The results and discussion section should be separate. Please improve the discussion section, add strong logical reasoning, and compare the results with previous studies.
Reply: The “results and discussion” section has been divided, and the discussion has been modified as suggested by the reviewer.
- The authors have collected only limited data, which is not good enough to publish in high-quality journals like Plants. I suggest adding more data on physiological and biochemical traits.
Reply: We understand the reviewer’s point of view. Currently, we are conducting further experiments in planta (starting with sowing seeds) to confirm the data obtained from our explants. However, please note that woody species require a longer growth period.
- The conclusions are not quantitative and should be rewritten for clarity, conciseness, and improved English.
Reply: The conclusions in the revised version have been carefully rewritten as suggested.
Round 2
Reviewer 1 Report
Comments and Suggestions for Authors
I saw a shift in approach to writing the article, and the end message is more clear and structured. This plant's potential for phytoremediation is more clearer. Anyway, as you mentioned, it is critical to evaluate a complete plant to determine the true potential.
Author Response
I saw a shift in approach to writing the article, and the end message is more clear and structured. This plant's potential for phytoremediation is more clearer. Anyway, as you mentioned, it is critical to evaluate a complete plant to determine the true potential.
Reply: We appreciate the reviewer’s comments.
Reviewer 2 Report
Comments and Suggestions for Authors
The authors have significantly improved the presentation of the manuscript; however, some points still need to be addressed before final publication.
1. What was the hypothesis of this study? It should be added in the last paragraph of the introduction section.
2. The presentation should be improved by adding correlation analysis.
Good luck
Author Response
The authors have significantly improved the presentation of the manuscript; however, some points still need to be addressed before final publication.
1. What was the hypothesis of this study? It should be added in the last paragraph of the introduction section.
Reply: The study’s hypothesis was stated in the last paragraph of the revised manuscript in the last submission as suggested “…. The ability of T. brasiliensis to thrive in acidic and metal-rich soils led us to hypothesize that this woody species might evolve robust metal tolerance mechanisms. Therefore, the aim of this study…”
- The presentation should be improved by adding correlation analysis.
Good luck
Reply: The correlation analysis was added as suggested by the reviewer. The analysis was included as a supplementary table in the revised version.
Reviewer 3 Report
Comments and Suggestions for Authors
The manuscript can be forward for publication.
Author Response
The manuscript can be forward for publication.
Reply: We appreciate the reviewer’s decision.